



# Actuator line model using simplified force calculation methods

Gonzalo Pablo Navarro Diaz[1], Alejandro Daniel Otero[2,4], Henrik Asmuth[1], Jens Nørkær Sørensen[3], and Stefan Ivanell[1]

[1]Uppsala University, Wind Energy Section, Campus Gotland, Visby, Sweden.
[2]Universidad de Buenos Aires, Facultad de Ingeniería, Buenos Aires, Argentina.
[3]Technical University of Denmark.
[4]Computational Simulation Center (CSC-CONICET), Buenos Aires, Argentina..

**Correspondence:** Gonzalo Pablo Navarro Diaz (gonzalopablo.navarrodiaz@vattenfall.de)

**Abstract.** To simulate transient wind turbine wake interaction problems using limited wind turbine data, two new variants of the actuator line technique are proposed in which the rotor blade forces are computed locally using generic load data. The proposed models, which are extensions of the actuator disc force models proposed by Navarro Diaz et al. (2019a) and Sørensen et al. (2020), only demand thrust and power coefficients and the tip speed ratio as input parameters. In the paper it is shown

the analogy between the Actuator Disc Method (ADM) and the Actuator Line Method (ALM) and from this derive a simple methodology to implement local forces in the ALM without the need for knowledge of blade geometry and local airfoil data. Two simplified variants of ALMs are proposed, an analytical one based on Sørensen et al. (2020) and a numerical one based on Navarro Diaz et al. (2019a). The proposed models are compared to the ADM, using analogous data, as well as to the classical ALM based on blade element theory, which provides more detailed force distributions by using airfoil data. To evaluate the

local force calculation, the analysis of a partial wake interaction case between two wind turbines is carried out for a uniform laminar inflow and for a turbulent neutral atmospheric boundary layer inflow. The computations are performed using the Large Eddy Simulation facility in OpenFOAM, including SOWFA libraries and the reference NREL 5MW wind turbine as test case. In the single turbine case, computed normal and tangential force distributions along the blade showed a very good agreement between the employed models. The two new ALMs exhibited the same distribution as the ALM based on geometry and airfoil

data, with minor differences due to the particular tip correction needed in the ALM. For the challenging partially impacted wake case, the analytical and numerical approaches manage both to correctly capture the force distribution at the different regions of the rotor area, with, however, a consistent overestimation of the normal force outside the wake and an underestimation inside the wake. The analytical approach shows a slightly better performance in wake impact cases compared to the numerical one. As expected, the ALMs gave a much more detailed prediction of the higher frequency power output fluctuations than the ADM.

These promising findings open the possibility to simulate commercial wind farms in transient inflows using ALM, without having to get access to actual wind turbine and airfoil data, which in most cases are restricted due to confidentiality.





# 1 Introduction

In recent years, Large Eddy Simulation (LES) of wind farms has become feasible due the emergence of fast computers and the development of the actuator disc and lines methods. Hence, today it is possible to capture the interaction of the wind flow
between the wind turbines (WTs) and transient problems in the atmospheric boundary layer (ABL), including the variation of the mean wind and turbulence associated with the diurnal cycle (Abkar et al., 2016; Englberger and Dörnbrack, 2018), as well as ramps and changes of wind directions (Arthur et al., 2020; Stieren et al., 2021). Using LES techniques facilitate the understanding of the interaction between the ABL and the WTs in wind farms, and makes it possible to capture the variation in the structure of WT wakes and their effects on loads and power losses (Abkar et al., 2016; Porté-Agel et al., 2020). The
study of the variation of WT power output for different inflow conditions is also important for determining the necessary fill-in power needed to follow a signal from the grid operator (Bossuyt et al., 2016). The well-known excessive computational cost that constrains the usability of LES based on Navier Stokes equations implementations on central processing units (CPUs) has been drastically reduced due to the new implementations in graphics processing units (GPUs), for example using the Lattice Boltzmann method (Asmuth et al., 2022), making LES studies accessible for large wind farms and in long time periods. One
of the challenges when simulating transient WT wake problems is the need for access to detailed wind turbine data, such as the operating setting and the geometric and aerodynamic characteristics of the blade. This information, which usually is subject to confidentiality, is needed for the calculation of blade forces using the classical blade element (BE) theory, which is an inherent part of the actuator line model (ALM) (Sørensen and Shen, 2002). In the ALM, the forces are distributed over rotating lines representing the effect of the blades on the flow, which enables transient simulations of the flow in wind farms. In most cases,
however, it is only possible to get access to generic open WT data, i.e. rotor radius ($R$), and thrust and power coefficients ($C_T$ and $C_P$) as function of tip speed ratio ($\lambda$).

This kind of input data is normally sufficient for performing computations using the Actuator Disc Method (ADM), as the loading in this method is spread over the surface (the disc) representing the swept rotor area (Navarro Diaz et al., 2019a). However, distributing the forces on a disc, instead of on rotating lines, does not capture correctly the physics of the vortex
pattern in the wake. A LES study by Martinez et al. (2012) showed that the root- and tip-vortices produced by the blades in the ALM are not developed in the ADM, which only generates a shear layer surrounding the wake. A similar LES comparison was carried out by Stevens et al. (2018), who found differences between the two models up to 3 rotor diameters downstream. Furthermore, (Marjanovic et al., 2017) carried out ALM computations in realistic ABL flows and found that the ALM generates distinct tip and root vortices that keep their coherence as helical tubes at least one diameter downstream of the rotor plane. Seen
in light of the possibility of using generic load distributions in the ADM and the advantage of the ALM to capture correctly transient flows, we here explore the possibility of combining the two methods by implementing a generic load model in the ALM.

In order to develop a generic load model for the ALM, we exploit the recent achievements of the ADM to improve the calculation of forces from local information of the flow over the disc, while keeping the input parameters on a minimum. Using
local flow properties allows the model to have a better performance in complex inflow conditions, such as ABL profiles in flat





or complex terrain, or for rotors subject to partial or total wake impact. The first example is the analytical ADM developed by Sørensen et al. (2020), in which analytical expressions specify the radial distribution for both the normal and tangential components of the blade body force. This formulation assumes that the rotor disc is subject to a constant circulation corrected for tip and root effects, and has the advantage that it for a given wind turbine size only demands the thrust and power coefficients

as a function of tip speed ratio as input. The calculation of the forces is done locally, hence the model is capable of handling non-uniform velocity fields over the rotor, for example produced by the vertical velocity profile of the ABL or the impact of wakes from surrounding wind turbines. The model was recently extended to determine the annual energy production in small and large wind farms by van der Laan et al. (2022). Another example is the model by Navarro Diaz et al. (2019b), where the local adaptation of the ADM is obtained by using a numerical approach. This procedure consists of deriving two pre-calibrated

tables for a wide range of WT operating conditions, which subsequently are used for the calculation of the local forces on the rotor plane. The analytical and numerical approaches perform very well for velocities below rated power, as the authors explain in their original papers (Sørensen et al., 2020; Navarro Diaz et al., 2019a). For velocities above the rated, however, the two approaches still need some improvements.

The novelty of the present work is the development of two new ALMs which, based on the earlier developed analytical and

numerical ADM approaches, are capable of simulating transient WT wake flows without the need of blade geometry and local airfoil data. The models are verified and compared for a wide range of flow cases, covering both uniform inflow and partial wake impact subject to non-turbulent as well as ABL turbulent ambient conditions. Furthermore, the capabilities of the ALMs to improve time dependent response of the turbine are studied.

This work is organized as follows. First, in Sect. 2.1 the similarities and differences between the ADM and ALM concepts

are described, followed by the formulation for the local force calculation based on three different approaches; one requiring detailed airfoil data as input (Sect. 2.1.1) and the two others only requiring the thrust and power coefficients as input parameters (Sect.s 2.1.2 and 2.1.3). The OpenFOAM software configuration is described in Sect. 2.2, followed by the description of the NREL-5MW reference WT (Sect. 2.3). The models are tested in Sect. 3 for different inflow conditions; first, for a uniform non-turbulent inflow with a single (Sect. 3.1.1) and two not-aligned WTs layout (Sect. 3.1.2). Second, for a turbulent ABL

inflow with two non-aligned WTs (Sect. 3.2). Finally, in Sect. 4 conclusions are presented.

## 2 Methodology

### 2.1 Force modeling

The basic idea behind the actuator disc is to replace the 3D geometry of the blades by equivalent forces acting on the flow. This strategy allows to save computational costs, avoiding the need to resolve the blade boundary layer and replacing the otherwise

boundary-fitted mesh around the solid blade surface by a coarse mesh with distributed volumetric forces (Sanderse et al., 2011). Fig. 1 depicts the two most common actuator WT models. First, in the actuator line model, ALM, the WT is represented by rotating lines moving with a rotational speed, $\omega$. In this model, the number of lines, $n_l$, is equal to the number of blades, $B$. These lines are discretized in nodes, $i$, which represent the contribution of a radial fragment of the blade, $\Delta_b$, where the normal,

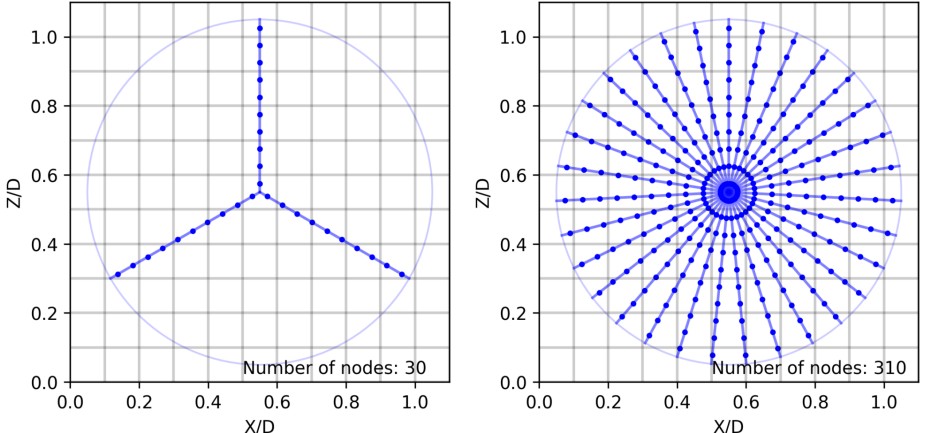

**Figure 1.** Example of the nodes distribution strategy on lines for the ALM (left) and ADM (right). In this example, a coarse mesh of 10 cells along the diameter is taken to make the node distribution clearer for the reader.

$F_{n,i}$, and tangential, $F_{\theta,i}$, forces are calculated. The separation between the nodes $\Delta_b$ is recommended to be dependent on the
horizontal cell size, $\Delta_x$, assuming $\Delta_b = 0.5\Delta_x$ (Asmuth et al., 2020). In the actuator disc model, ADM, the forces are instead
distributed over the entire rotor swept area. To make an analogy to the ALM, in this work the disc is represented by multiple
non-rotating artificial lines (Nathan et al., 2015; Martinez et al., 2012). The idea behind the proposed technique is that there
exists a full correspondence between the ADM and the ALM method, such that the advantage of using generic WT data in the
ADM also can be exploited in the ALM method. To ensure that there is at least one node in each cell at the tip of the ADM,
the number of lines $n_l$ in the ADM is calculated as:

$$n_l = \frac{2\pi R}{\Delta_x}, \tag{1}$$

where $R$ is the radius of the rotor blade. In the ADM, using the described method, for each time step in the solution procedure
the forces are computed and distributed on all nodes on the entire surface disc. In contrast to this, in the ALM only the nodes
representing the blades are active as carriers of the forces. Due to the rotation of the blades, the active points move as a function
of time. Using a time step

$$\Delta t = \frac{\Delta\theta}{\omega} = \frac{2\pi}{\omega n_l}, \tag{2}$$

ensures that the blade tip does not translate faster than one cell per time step (see Fig. 1). One important characteristic to
mention is the amount of nodes needed in each model, since this quantity has a direct relation with the computational cost of
the model for each time step. Using the analogy between ADM and ALM presented here, the ADM seems to demand 10 times
more nodes than the ALM. However, in reality much fewer points are needed in an actual computation using ADM and the
ALM will generally be more computing demanding due to the limitation of the time step.

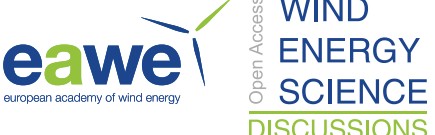

In both models, ALM and ADM, the normal ($\Delta F_{n,i}$) and tangential ($\Delta F_{\theta,i}$) forces per area unit are calculated based on the force's calculation method adopted and then concentrated ($F_{n,i}$ and $F_{\theta,i}$) for each node $i$ as

$$F_{n,i} = \Delta F_{n,i} \Delta A_i, \tag{3}$$


$$F_{\theta,i} = \Delta F_{\theta,i} \Delta A_i, \tag{4}$$

where $\Delta A_i$ is the disc area corresponding to each node, obtained as

$$\Delta A_i = \frac{2\pi r_i \Delta_b}{n_l}, \tag{5}$$

and $r_i$ the local radial position from the center of the turbine.

It should be noted that, in this work, the only difference between the ALM and the ADM is the number of lines $n_l$ and the tip and root corrections adopted. To express the normal and tangential forces per length unit along the blade ($f_{n,i}$ and $f_{\theta,i}$), node forces are divided by the blade element length $\Delta_b$ as:

$$f_{n,i} = \frac{F_{n,i}}{\Delta_b}, \tag{6}$$

$$f_{\theta,i} = \frac{F_{\theta,i}}{\Delta_b}. \tag{7}$$

Finally, the total thrust, $T$, and power, $P$, are obtained as

$$T = \sum_i F_{n,i}, \tag{8}$$

$$P = \omega \sum_i F_{\theta,i} r. \tag{9}$$

To avoid numerical instability, the normal and tangential forces in each node are then distributed in by means of a three-dimensional Gaussian function using a regularization kernel ($\eta_{cell}$) (Porté-Agel et al., 2011) in neighbor cells,

$$F_{n,cell} = F_{n,i} \frac{\eta_{cell} V_{cell}}{\sum_{cell} \eta_{cell} V_{cell}}, \tag{10}$$

$$F_{\theta,cell} = F_{\theta,i} \frac{\eta_{cell} V_{cell}}{\sum_{cell} \eta_{cell} V_{cell}}, \tag{11}$$


$$\eta_{cell} = \frac{1}{\varepsilon^3 \pi^{3/2}} \exp\left[-\left(\frac{d}{\varepsilon}\right)^2\right], \tag{12}$$





**Table 1.** Actuator disc and actuator line models employed in this work, describing the differences on the local force calculation and the chosen tip and root corrections.

| Actuator model | Name | Local force | Root correction | Tip correction |
|---|---|---|---|---|
| Actuator disc | ADM-airfoil | Airfoil based | none | Glauert (Glauert, 1935) |
| | ADM-an | Analytical | Vortex core model (Sørensen et al., 2020) | Glauert |
| | ADM-num | Numerical | Vortex core model | Glauert |
| Actuator line | ALM-airfoil | Airfoil based | none | Variable $\varepsilon$ (Dağ and Sørensen, 2020) |
| | ALM-an | Analytical | Vortex core model | Variable $\varepsilon$ |
| | ALM-num | Numerical | Vortex core model | Variable $\varepsilon$ |

where $d$ is the distance between the node and the center of the cell and $\varepsilon$ is the smearing factor, commonly fixed as $\varepsilon = 2\Delta_x$ (Howland et al., 2016). To take into account this force distribution in non-uniform meshes, the volume of the cell ($V_{cell}$) is considered in the force Eq. 10 and 11.

WTs in wind farms are affected by non-uniform velocity fields due to the ABL flow, the influence of the topography and the upstream WT wakes. This creates the necessity of a local force calculation in the ALM and ADM in order for the model to give a better and more realistic response to this complex velocity field. The local calculation of $\Delta F_{n,i}$ and $\Delta F_{\theta,i}$ depends on the available information of the WT. If the operating setting and the geometrical and aerodynamic characteristics of the blade are available, the classically BE theory can be used. On the other hand, if only generic WT data, i.e. $R$, $C_T$, $C_P$ and $\lambda$, are

available then the forces can be calculated by means of the analytical (Sørensen et al., 2020) or numerical (Navarro Diaz et al., 2019b) approach.

     Based on the two types of actuator models, ALM and ADM, and the three approaches to calculate the local forces, BE, analytical and numerical, six models in total are implemented. Table 1 shows an overview of the different models, specifying the tip and root corrections as well as the naming convention. In the following, the calculation procedure of the local forces is

explained for the various models.

### 2.1.1    Airfoil based forces

The first ADM model combining blade element (BE) theory with Computational Fluid Dynamics (CFD) is due to Sørensen and Myken (1992); Sørensen and Kock (1995). This model was later extended to ALM and used in combination with LES (Sørensen and Shen, 2002; Nathan et al., 2018; Sørensen et al., 2015; Nathan et al., 2017; Asmuth et al., 2020). Later developments

combining LES with ADM are e.g. due to Porté-Agel et al. (2011); van der Laan et al. (2015). Using a BE approach requires detailed knowledge regarding the WT operating setting as well as the geometrical and aerodynamic characteristics of the rotor blade.

     First, the general operating regime of the WT needs to be obtained, which is related to the reference velocity $U_{ref}$. Knowing $U_{ref}$ the pitch angle $\Theta_p$ and rotational speed $\omega$ can be extracted from the WT characteristics. In case $U_{ref}$ is unknown, a

torque controller is the standard method that provides the rotational speed of the turbine. In cases when a torque controller is





not possible to implement due to the lack of blade information, a simple calibration table can be used (van der Laan et al., 2015) by creating a calibration table with the relation between the average velocity over the disc ($\langle U_d \rangle$) and the $U_{ref}$. This approach can not account for inertial affects on the rotational speed of the rotor.

Once the operating regime is determined, the forces are calculated for each node, as described in the following. First, the relative velocity $U_{rel}$ in the blade element is obtained as

$$U_{rel} = \sqrt{U_n^2 + (U_\theta - \omega r)^2}, \tag{13}$$

where $U_n$ and $U_\theta$ are the local normal and tangential velocities, respectively. The angle between $U_{rel}$ and the rotor plane is calculated as

$$\Phi = \arctan\left[\frac{U_n}{U_\theta - \omega r}\right]. \tag{14}$$

The local angle of attack is determined as $\alpha = \Phi - \beta$, where $\beta$ is the sum of the flow angle $\Theta_p$ and the *twist* of the blade in the local radial position $r$. Then, the lift, $\Delta F_L$, and drag, $\Delta F_D$, forces per unit area are obtained as

$$\Delta F_L = \frac{1}{2}\rho U_{rel}^2 \frac{Bc}{2\pi r}\mathcal{F}_{tip}C_L e_L, \tag{15}$$

$$\Delta F_D = \frac{1}{2}\rho U_{rel}^2 \frac{Bc}{2\pi r}\mathcal{F}_{tip}C_D e_D, \tag{16}$$

where the airfoil information needed are the chord length, $c$, the lift and drag coefficients, $C_L$ and $C_D$, and the lift and drag unit vectors, $e_L$ and $e_D$. In these equations $\rho$ is the air density, $B$ is the number of blades ($B = 3$) and $\mathcal{F}_{tip}$ and $\mathcal{F}_{root}$ are the tip and root corrections, respectively. The corrections used in this work are separately explained in Sect. 2.1.4. Finally, the normal and tangential components per unit of area are obtained as

$$\Delta F_n = \Delta F_L \cos\Phi + \Delta F_D \sin\Phi, \tag{17}$$


$$\Delta F_\theta = \Delta F_L \sin\Phi - \Delta F_D \cos\Phi. \tag{18}$$

### 2.1.2   Analytical force distribution

The analytical local forces calculation was originally proposed by Sørensen et al. (2020) to be used in the ADM, analysing the performance of this new model for both a single and two aligned WTs in a ABL inflow. Also, this model was validated in 180 (Sørensen and Andersen, 2020) for four different WTs operating under a wide range of conditions. The main advantage of this analytical ADM is that the expressions only depend on global parameters, such as rotor radius $R$, tip speed ratio $\lambda$, and thrust coefficient $C_T$.





In this work, the implementation of this model is done following the next steps. First, in the cases where $U_{ref}$ is not fixed and needs to be estimated, the average velocity over the rotor disc $\langle U_d \rangle$ is determined and the WT operating regime is obtained by determining $U_{ref}$ using the analytical formulation

$$U_{ref} = \frac{2\langle U_d \rangle}{1 + \sqrt{1 - C_T}}, \tag{19}$$

which is based on the same relation present in Eq. 26. Due to the fact that commercial WTs, as well as the reference turbine used in this work, have a $C_T$ curve depending on $U_{ref}$, an iterative process of calculating $U_{ref}$ is needed. Next, the tip speed ratio, $\lambda$, is calculated as

$$\lambda = \frac{\omega R}{U_{ref}}. \tag{20}$$

This model formulation requires the tip $\mathcal{F}_{tip}$ and root $\mathcal{F}_{root}$ corrections for all the radial position. Thus the dimensionless reference circulation, $q_0$, is calculated as

$$q_0 = \frac{\sqrt{16\lambda^2 a_2^2 + 8a_1 C_T} - 4\lambda a_2}{4a_1}, \tag{21}$$

where $a_1$ and $a_2$ are the result of integrating $\mathcal{F}_{tip}$ and $\mathcal{F}_{root}$ along the blade as

$$a_1 = \int_0^1 \frac{\mathcal{F}_{root}^2 \mathcal{F}_{tip}^2}{x} dx, \tag{22}$$

$$a_2 = \int_0^1 \mathcal{F}_{root} \mathcal{F}_{tip} x \, dx, \tag{23}$$

where $x = r/R$ is the non dimensional radial distance, depending on the local radius $r$. Finally, the local normal, $\Delta F_{n,i}$, and tangential, $\Delta F_{\theta,i}$, forces over the disc area for each node $i$ are calculated as

$$\Delta F_{n,i} = \rho q_0 \frac{\mathcal{F}_{root} \mathcal{F}_{tip}}{x} \left( \lambda x + \frac{1}{2} q_0 \frac{\mathcal{F}_{root} \mathcal{F}_{tip}}{x} \right) U_\infty^2, \tag{24}$$

$$\Delta F_{\theta,i} = \rho q_0 \frac{\mathcal{F}_{root} \mathcal{F}_{tip}}{x} U_\infty^2 \frac{\left( 1 + \sqrt{1 - C_T} \right)}{2}. \tag{25}$$

In these two equations $U_\infty$ is the upstream unperturbed velocity for forces calculation, which is defined in the analytical ADM as:

$$U_\infty = \frac{2U_{d,i}}{\left( 1 + \sqrt{1 - C_T} \right)}, \tag{26}$$

where $U_{d,i}$ is the local disc velocity at the node.





### 2.1.3 Numerical force distribution

A local calculation of forces using a numerical approach was originally proposed by Navarro Diaz et al. (2019b) to improve
the performance of the ADM for complex flows in wind farm simulations. This methodology was also used in four ADMs
with increasing levels of complexity (Navarro Diaz et al., 2019a), proving that the local calculation improves the solution for
wake interaction cases. The model has shown a satisfactory performance for capacity factor estimation in an onshore wind
farm (Navarro Diaz et al., 2021).

In the formulation from the original publication, with no tip and root corrections included, the non-corrected normal $\Delta F'_n$
and tangential $\Delta F'_\theta$ forces per unit of area are calculated as

$$\Delta F'_{n,i} = \frac{1}{2}\rho C_T U_\infty^2, \tag{27}$$

$$\Delta F'_{\theta,i} = \frac{\rho C_P U_\infty^3}{2\omega r}. \tag{28}$$

In this numerical adaptation approach, $U_\infty$ is the upstream velocity which, for uniform incoming flows, can be assumed
equivalent to $U_{ref}$.

In order to apply the tip $\mathcal{F}_{tip}$ and root $\mathcal{F}_{root}$ corrections and conserve the total thrust and power output, the normal $\Delta F_n$ and
tangential $\Delta F_\theta$ forces per unit of area are scaled as

$$\Delta F_{n,i} = b_1 \mathcal{F}_{tip}\mathcal{F}_{root}\Delta F'_{n,i}, \tag{29}$$

$$\Delta F_{\theta,i} = b_2 \mathcal{F}_{tip}\mathcal{F}_{root}\Delta F'_{\theta,i}, \tag{30}$$

where $b_1$ and $b_2$ are scale factors for the normal and tangential forces, respectively, obtained as

$$b_1 = \frac{\sum_i F'_{n,i}}{\sum_i F'_{n,i}\mathcal{F}_{tip}\mathcal{F}_{root}}, \tag{31}$$

$$b_2 = \frac{\sum_i F'_{\theta,i} r}{\sum_i F'_{\theta,i} r \mathcal{F}_{tip}\mathcal{F}_{root}}. \tag{32}$$

In the cases when the upstream reference wind velocity $U_{ref}$, which defines the turbine setting given by $C_P$, $C_T$ and $\omega$, is
unknown, it needs to be estimated from the average wind velocity over the disc area $\langle U_d \rangle$. Note that the wind velocity at a
certain location on the disc $U_{d,i}$ does not necessarily have to coincide with $\langle U_d \rangle$, especially under complex flow patterns. This
$U_{d,i}$ is the velocity that defines the local effect of the actuator disc on the flow. So, given a particular turbine setting defined
by $U_{ref}$, at each location there will be a relation between $U_{d,i}$ and a corresponding unperturbed upstream wind velocity $U_\infty$
which will determine the local forces. This differentiation between the wind velocity adopted in the WT setting $U_{ref}$ and the



**Table 2.** Example of the table defining the relation between the average velocity on the disc ($\langle U_d \rangle$) and the reference operational velocity of the WT ($U_{ref}$), with the corresponding coefficients $C_T$ and $C_P$. The values were calculated using the ADM-airfoil and corresponds to inflow velocities below and above rated velocity ($U_{ref} = 11.4\ \mathrm{ms^{-1}}$) for the NREL-5MW.

| $U_{ref}$ | $C_T$ | $C_P$ | $\langle U_d \rangle$ |
|---|---|---|---|
| 11 | 0.751 | 0.535 | measured |
| 12 | 0.717 | 0.518 | measured |
| ... | ... | ... | ... |

**Table 3.** Example of the calibration table to obtain the relations between the local velocity on the node ($U_{d,i}$) and the unperturbed velocity ($U_\infty$) used for the local forces calculation on the node.

| $U_{ref}$ | $C_T$ | $C_P$ | $U_\infty$ | $U_{d,i}$ | $\Delta F_{n,i}$ | $\Delta F_{\theta,i}$ |
|---|---|---|---|---|---|---|
| 11 | 0.751 | 0.535 | 4 | measured | calculated | calculated |
|  |  |  | 5 | measured | calculated | calculated |
|  |  |  | ... | ... | ... | ... |
| 12 | 0.717 | 0.518 | 4 | measured | calculated | calculated |
|  |  |  | 5 | measured | calculated | calculated |
|  |  |  | ... | ... | ... | ... |
| ... | ... | ... | ... | ... | ... | ... |

unperturbed wind velocity used to calculate the ADM forces $U_\infty$, introduced by Navarro Diaz et al. (2019a), provides the ADM with the capability to adapt the forces in areas that work under off-design conditions.

The numerical ADM implementation requires a calibration process where the turbine is operated to face different uniform inflow cases. This process allows to establish a relationship between local velocities at the ADM and the corresponding unperturbed velocities: firstly, the relation between $\langle U_d \rangle$ and $U_{ref}$ (defining the respective $C_T$ and $C_P$); and secondly for each $U_{ref}$
configuration, the relation between $U_{d,i}$ and $\Delta F_{n,i}$ and $\Delta F_{\theta,i}$. These relations are expressed in two tables: 2 and 3. To construct table 2, a simulation is run for each inflow velocity $U_\infty$ in the WT operating range, making the ADM impose the forces $\Delta F_{n,i}$ and $\Delta F_{\theta,i}$ on the fluid according to the thrust and power at that given velocity. Thus, the resulting average velocity $\langle U_d \rangle$ is measured, disclosing the relation between $\langle U_d \rangle$ and the reference velocity $U_{ref}$ assumed equal to the $U_\infty$ in that case, in a procedure similar to that proposed by van der Laan et al. (2015). The second and third columns of table 2 come from the WT
manufacturer specifications.

To construct table 3, for a each $C_T$, $C_P$ and $\lambda$ in accordance to a reference velocity $U_{ref}$ in the WT operating range, different inflow velocities $U_\infty$ are imposed in each simulation along with the resulting ADM forces, saving for each discrete radial position $r_i$, the local velocity $U_{d,i}$, and forces $\Delta F_{n,i}$ and $\Delta F_{\theta,i}$, columns 5 to 7 of table 3. Compared to the procedure followed in (Navarro Diaz et al., 2019a) where there was no tip and root correction, the values of $\Delta F_{n,i}$ and $\Delta F_{\theta,i}$ are needed
in this case as they include these corrections.

Afterwards, in each time step of the wind farm simulation, the ADM implementation takes the average velocity over the disc $\langle U_d \rangle$, and uses table 2 to interpolate $U_{ref}$ and obtain the $C_T$ and $C_P$ values. Then, in each ADM node, it takes the velocity $U_{d,i}$ and by means of table 3 interpolates the corresponding $\Delta F_n$ and $\Delta F_\theta$.



### 2.1.4 Tip and root corrections

The tip and root corrections adopted in each ADM and ALM are summarized in Table 1. In the case of the ADM, this model assumes that the flow is azimuthally uniform in the rotor plane, and thus, requires a tip loss correction to take into account the effect of a finite number of blades. For this work, the Glauert tip loss correction (Glauert, 1935) is used, following the recommendation by Sørensen and Andersen (2020). This correction was developed from the study of Prandtl (Zhong et al., 2019) and can be expressed as

$$\mathcal{F}_{tip} = \frac{2}{\pi} \arccos \left[ \exp \left( -\frac{B(1-x)}{2x \sin \Phi} \right) \right]. \tag{33}$$

In the ALM there is also the need of introducing another type of correction to address the problem of the thickness of the shed vortices in ALM produced by smearing the blade forces into the CFD domain. This has lately been discussed and solved in the works by Jha and Schmitz (2018) and Dağ and Sørensen (2020). The correction proposed by Jha and Schmitz (2018) is the most suitable for this work, due to the minor blade information needed in the formulation. This correction consists of

making $\epsilon$ dependent on the local radial position, with the restricting it to $\epsilon = 1$ at the tip. The only limitation on this correction is the need of the chord length values at each airfoil section. To overcome this, a simpler $\epsilon$ variation is proposed, setting $\epsilon = 2$ from the root up to the middle of the blade, and then reducing it linearly to $\epsilon = 1$ at the tip.

The root correction $\mathcal{F}_{root}$ is based on a vortex core model, used in the original work of the ADM-an (Sørensen et al., 2020), to account for the inner non-lifting part of the rotor. This part of the blade is only correctly described in the ADM-airfoil and

ALM-airfoil models avoiding the need of root correction. For the rest of the models, the correction is expressed as

$$\mathcal{F}_{root} = 1 - \exp \left[ -a \left( \frac{x}{\overline{\delta}} \right)^b \right], \tag{34}$$

where $b = 4$ and $a = 2.335$ are two coefficients, which values are adopted from the original analytical ADM paper. $\overline{\delta} = \delta/R$ is the vortex core, defined as the non-dimensional distance from the root to the point where the lift force becomes active. This distance depends on the blade and, for the reference turbine used in this work, a value $\overline{\delta} = 0.07$ gives the best fit compared to

the ADM based on airfoil data.

### 2.2 Numerical set-up

In this work the open source software OpenFOAM (version 2.3.1) is used in conjunction with the SOWFA project libraries (Churchfield et al., 2012b) developed by the U.S. National Renewable Energy Laboratory (NREL). LES is chosen in order to capture the transient flow solution using ALM and compare it with the ADM. Both sets of turbine models, ALM and

ADM, have been re-implemented in comparison with the original code provided in SOWFA following the above-mentioned developments. This implementation strategy allows the ALM and ADM to have the same base code, meaning that the node distribution and tip and root corrections are the only sources of differences. A simple one-equation eddy viscosity approach is employed as Sub-Grid Scale (SGS) model (Yoshizawa, 1986). Two flow situations, uniform and ABL inflows, are considered to analyze the model performance for different ambient turbulence conditions.



Both single turbine and two turbine interaction configurations are performed for the uniform inflow case. In the single turbine case, two domains are used. For the mesh sensitivity analysis a cubic domain configuration with dimensions of 6D on each side is specified, with the turbine located at the center of the domain. For the model and wake comparison a longer domain in the stream-wise direction is used, consisting of a 10D long domain located downstream of the turbine position. The mesh consists of cubic cells, having different refinement zones near the turbine in order to achieve the desired resolution in the rotor disc.

The details of the refinement zones can be seen in Fig. 2. In the two turbine interaction case, the turbines (WT1 upstream and WT2 downstream) are 5D apart in the stream-wise direction and 0.5D in the span-wise direction. This layout generates a half wake impact on the rotor plane of the downstream turbine, WT2, representing one of the most complex flow conditions that a turbine model can face. This layout is similar to the one used in Navarro Diaz et al. (2019a). The turbine positions together with the mesh refinement zones can be seen in Fig. 2. The simulation is performed in a time span of 660 s, with the last 60 s, corresponding to 9.15 blade revolutions, sampled for analysis.

For the ABL inflow, a neutral stratified condition is simulated and the turbulent solution is achieved by means of a precursor simulation with periodic boundary conditions. In this stage, a horizontal domain extension of $l_x \times l_y = 24D \times 24D = 3.024 \mathrm{km} \times 3.024 \mathrm{km}$ and a total height of 1 km is chosen (Churchfield et al., 2012b). In this domain, a capping inversion starting from 700 m above ground is fixed. The inclusion of the capping inversion and the domain dimensions follow the recommendation by

Churchfield et al. (2012b), who remarks that the capping inversion in neutral conditions helps to slow the vertical growth of the boundary layer with time. In SOWFA, a desired horizontal average velocity and direction at hub height can be forced in every time step by adjusting the magnitude and direction of the driving pressure gradient. Because this is an idealized study case, the terrain is flat and the Coriolis force is not included in the simulations. Due to the fact that the Coriolis force is neglected, the average velocity vector does not veer with height and, if the velocity direction is fixed at 270°, very large streamwise-elongated

coherent structures would only have 3km to travel before entering again into the domain. This situation results problematic in the solution when periodic boundary conditions are chosen, as also addressed by Munters et al. (2016). In order to avoid this problem, a direction which is not aligned with the domain sides is forced, giving more distance for the coherent structures to travel before entering from the same inlet face position. This strategy was also used in SOWFA by Wang et al. (2018). If a maximum coherent structure spanwise dimension ($d_y$) similar to the ABL height ($d_y = l_y/4 = 750$ m) is assumed, the

inclination angle can be calculated as $\gamma = \arctan(d_y/l_x) = 14.03°$. With this inclination, the structures travel through a total distance of $l_x \cos(\gamma) = 12.46$ km (more than four times $l_x$). This configuration can be seen in Fig. 3 with the total distance remarked as a continuous line on the left image. In the precursor, the cells in the mesh have a horizontal dimension of 16 m (approximately D/8) and a vertical dimension that grows with height, starting with 2.5 m and finishing with 60 m at the top. The roughness length ($z_0$) is determined according to the desired turbulence intensity ($TI$) at hub height, defined as

$$TI(\%) = \frac{100\sqrt{\frac{2}{3}TKE}}{U_{hub}},\tag{35}$$

where $TKE$ is the turbulent kinetic energy also at hub height. The relation between this $TI$ and the standard one obtained with a cup anemometer for neutral ABL is $TI_{cup} = TI/0.8$ (van der Laan M. P.; et al., 2015). In this work, $z_0 = 0.1$ m is adopted, corresponding to $TI = 7.8$ % ($TI_{cup} = 9.75\%$) similar to the one used in the two WT interaction study on a onshore wind



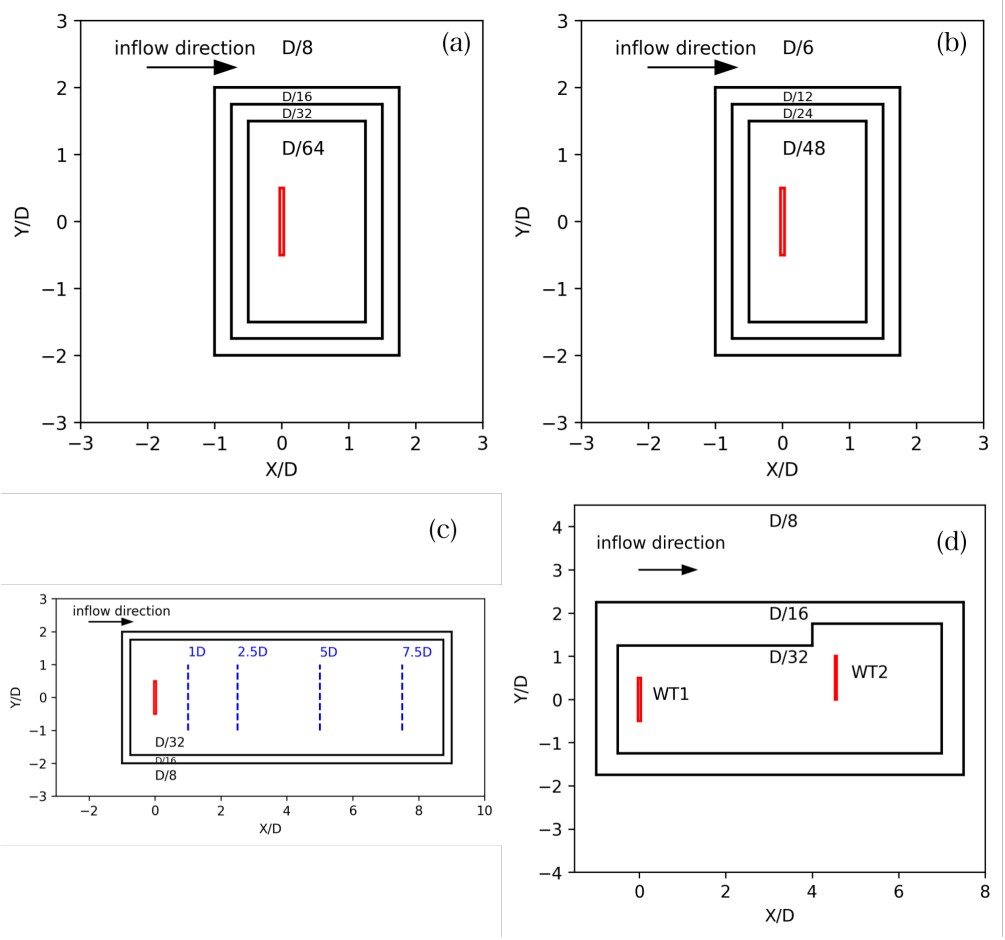

**Figure 2.** Horizontal domain size and cell size in the mesh refinement zones for the non-turbulent inflow cases: (a) and (b) for the mesh sensitivity analysis and (c) for the wake analysis with a single turbine. In (d) the domain for the two WTs layouts is shown. The only difference between the single turbine cases are the cell sizes.





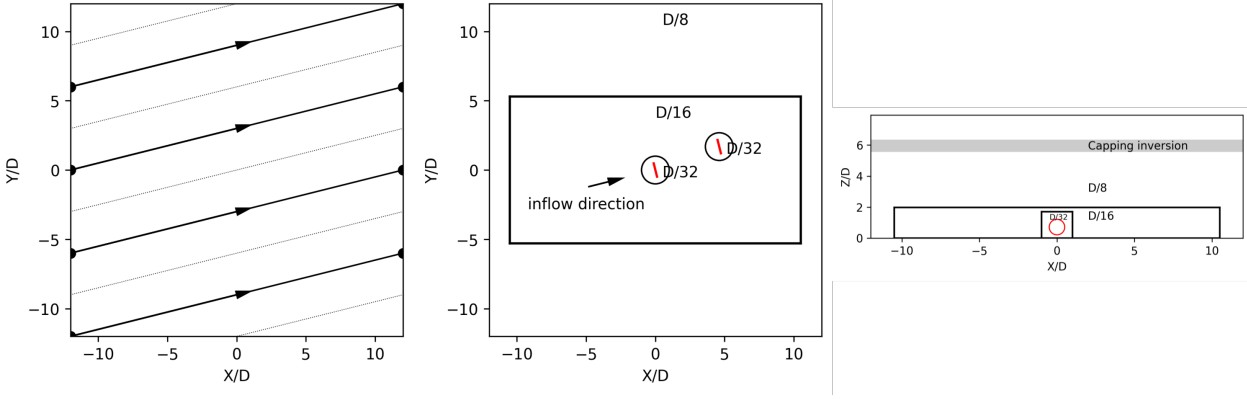

**Figure 3.** Domain size and mesh refinement zones for the ABL inflow case: (left) total distance traveled by the coherent structure before entering again into the domain in the same position and mesh refinement zones in a (center) horizontal and (right) vertical plane.

farm from van der Laan M. P.; et al. (2015). The precursor is computed for 5 hours (18000 s) to obtain a quasi-steady state
(Churchfield et al., 2012a). In the next hour and 10 min the values at the corresponding inlet faces are recorded for each time step, in order to use them for the farm.

In the farm stage, the mesh is refined in the wake zone and near the turbines. Fig. 3 shows more details of the domain size and mesh refinement zones in horizontal and vertical planes. It is important to remark that when the flow enters the refinement region of $D/16$ it travels a distance of 10.74D (1353m) before reaching the first turbine, a distance that is long enough to
develop the eddy structures in the new refined mesh. The cylindrical zone around each turbine is refined to $D/32$ with the objective to follow the mesh sensitivity study recommendation near the rotor plane (Sect. 3.1.1) without developing new eddy structures.

### 2.3 Reference wind turbine

In the present study, the widely employed NREL-5MW reference WT (Jonkman et al., 2009), which is equipped with a 3-
bladed rotor of diameter $D = 126$ m and tower height $H = 90$ m, is used for the computations. Regarding the WT operational configuration the turbine has two control regimes: one below the rated wind speed ($U_{ref} = 11.4 \,\mathrm{ms^{-1}}$) where $\omega$ grows linearly (with a $\lambda = 7.55$) and $\theta_p=0°$, and one above the rated wind speed where $\omega$ remains constant and $\theta_p$ increases. In contrast to the previous work (Navarro Diaz et al., 2019a), the $C_T$ and $C_P$ curves of the NREL-5MW are in the present work not taken from Ponta et al. (2016), but use the two curves resulting from the ADM-airfoil and ALM-airfoil. This ensures a fair comparison
between the actuator disc and actuator line model families.





# 3  Results

The results are separated between the uniform and turbulent ABL inflow condition. First, a single turbine is simulated with uniform inflow to compare all the variations of ADM and ALM proposed in this work. This comparison is done for a below rated inflow velocity $U_{hub} = 8 \text{ ms}^{-1}$. Once the models are tested in a single turbine setting, another step in the inflow complexity is achieved, locating two turbines in a not aligned layout, in order to test the models for a half wake impact situation on the rotor plane. This pair of turbines are located in uniform and turbulent ABL inflow conditions, with $U_{hub} = 8 \text{ ms}^{-1}$ at hub height. This difference in the turbulent condition will change the wake impact on the perturbed WT.

## 3.1  Uniform inflow

### 3.1.1  Single wind turbine

First, a preliminary mesh sensitivity study is necessary to find the dependency of the force´s distribution and the wake on the number of cells along the diameter. This pre-study can be found in the Sect. A in the Appendix. From the results obtained in this mesh sensitivity study, the resolution $D/32$ has been chosen for the simulations in this work. This choice aligns with recommendations from other authors for ALM simulation in uniform and turbulent ABL inflow (Asmuth et al., 2020, 2022).

Once the mesh resolution is chosen, the next step is the six models comparison for a basic case with a single turbine facing an uniform inflow velocity of $U_{hub} = 8 \text{ ms}^{-1}$, condition that determines $\Omega$, $C_T$ and $C_P$ but not the local force calculation. In order to obtain the $C_T$ and $C_P$ for the analytical and numerical models, the ADM-airfoil and ALM-airfoil are run for all the velocities $U_{ref}$ in the productive range with constant $\lambda = 7.55$ and the resulting values of $C_T$ and $C_P$ are saved in two tables, one for the ADMs group and another one for the ALMs group. The separation in two tables is required since the obtained coefficients are affected by the different tip and root corrections adopted in ADM-airfoil and in ALM-airfoil, respectively. These tables are used by the ADM-an, ADM-num, ALM-an and ALM-num. In Fig. 4 it can be seen the normal $f_n$ and tangential $f_\theta$ forces distribution, separating the results in two groups, ADMs and ALMs. In both groups of models, the major differences can be found in the tip and root regions while in the middle of the blade the constant circulation assumption matches very well with the calculations based on airfoil data. Specially for the tangential force distribution at the root, the models have large discrepancies due to the assumptions in the root correction compared to the use of airfoil data. It is important to notice that despite the use of tip correction in the ALM, there is a slightly over estimation of $f_n$ and $f_\theta$ in the last 10% of the blade.

In order to analyze how the differences in the force´s distribution affect the wake, in Fig. 5 the velocity magnitude is compared at four distances downstream the turbines, 1D, 2.5D, 5D and 7.5D. The wakes obtained with each model are remarkably close, with the major differences located in the central part of the rotor. As expected, the wakes tend to collapse on the same shape at the far wake region due to the use of the same $C_T$. The non-turbulent inflow condition increases the necessary distance for the wake results to coincide, reaching in this case more than 7.5D.

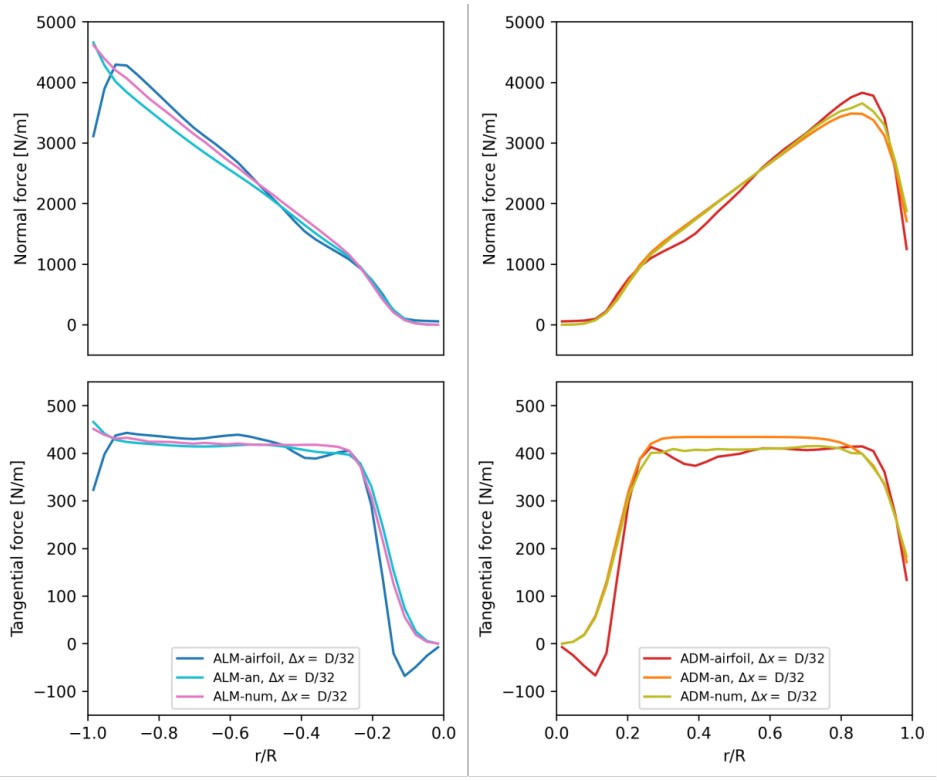

**Figure 4.** Model comparison for a single turbine case under uniform velocity inflow $U_\infty = 8$ ms$^{-1}$ and fixed $U_{ref} = 8$ ms$^{-1}$: normal $f_n$ (upper) and tangential $f_\theta$ (lower) forces distribution using different models.

### 3.1.2 Wind turbine interaction

The next step is to analyze how the models perform under non-uniform inflow conditions, in this case when WT1 generates a wake which afterwards affects WT2. Particularly for this work, the ADM-airfoil is always used to simulate WT1 in order to create the same wake inflow condition for WT2. Also, in WT2 the reference velocity $U_{ref}$ is fixed, a condition that determines

370 the rotation speed $\Omega$. In order to find the $U_{ref}$ for WT2, a preliminary simulation is carried out using ADM-airfoil also in the WT2, but in which $U_{ref}$ is estimated based on the average velocity on the disc $U_d$. In order to find a relation between $U_d$ and $U_{ref}$, a pre-calculated table is used. Because the estimated value of $U_{ref}$ for each time step is different, due to the non-uniform inflow, the value of $U_{ref}$ that will be used for in the next simulations is computed by time-averaging the values. Resulting in this case, $U_{ref} = 5.992$ ms$^{-1}$, with a corresponding $\Omega = 0.719$ $s^{-1}$.

375 Fig. 6 shows the instantaneous volumetric force and velocity on a vertical plane at WT2 position for the ADM-airfoil and ALM-airfoil. In the same Figure the velocity field on a horizontal plane at hub height is shown. In the region of wake impact on the WT2 rotor it can be noticed the local reduction of the forces, both in ADM-airfoil and ALM-airfoil, due to a lower inflow velocity. Especially for the ALM-airfoil, every time that the blade passes this wake region, the forces are instantaneously




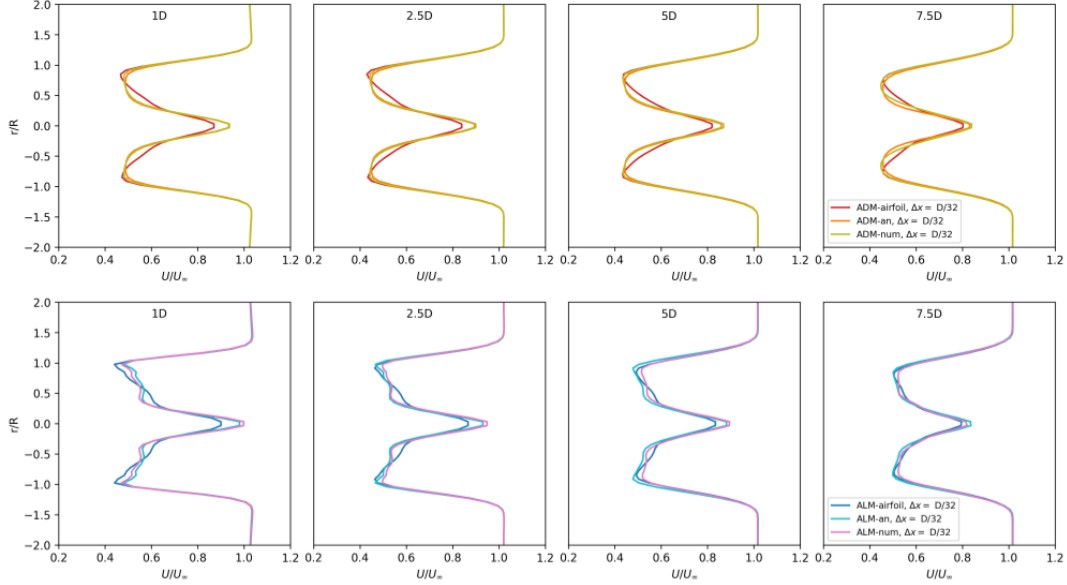

**Figure 5.** Velocity magnitude comparison on the wake using different models for single turbine in an uniform velocity inflow $U_\infty = 8$ ms$^{-1}$: ADM (top) and ALM (bottom).

reduced. For this transient problem, the period $T$ is deduced as the time that the blade in WT2 ($\Omega = 0.719$ $s^{-1}$) takes to move
120° ($2\pi/3$), $T = 2\pi/(3\Omega) = 2.9$ s. In order to see how this transient problem affects the turbine power output, in Fig. 7 the
power along 4 periods (11.6 s) using the three ALM is shown. As expected, the ADM-airfoil cannot capture the variation in
power and only reports a constant value (not shown in the figure), while all the ALM show the same pattern when a blade
enters and gets out the wake. Taking ALM-airfoil as the reference, the normalised power amplitude is 0.28. ALM-airfoil is the
model that reports the higher amplitude, followed by ALM-num and ALM-an.

In order to study how the models distribute the forces in this uneven spatial inflow, in Figure 8 the average normal $f_n$ and
tangential $f_\theta$ forces distribution on WT2 rotor, both for the impacted and free side are shown. When the ADMs are compared,
the ADM-an and ADM-num have a similar response at each side of the rotor, finding minor differences not only in the waked
side, but also for the free side. Compared to the more realistic ADM-airfoil, the tendencies in both sides are similar, with major
differences in the free side. When the three ALMs are compared, similar observations can be obtained. It is important to notice
that the overestimation of the normal force in the free side by the numeric approach is counteracted by an underestimation of
the tangential force in the same rotor area. The major difference in the free side explains the reason why the peaks in the power
output time series correspond to the moments when the models show the greatest differences. It can be noticed that the uniform
laminar inflow induces a wake impact with strong local velocity deficit on the rotor. This complex condition makes it difficult
for simple analytical and numerical approaches to have satisfactory performance in the entire rotor area.



WIND
ENERGY
SCIENCE
DISCUSSIONS

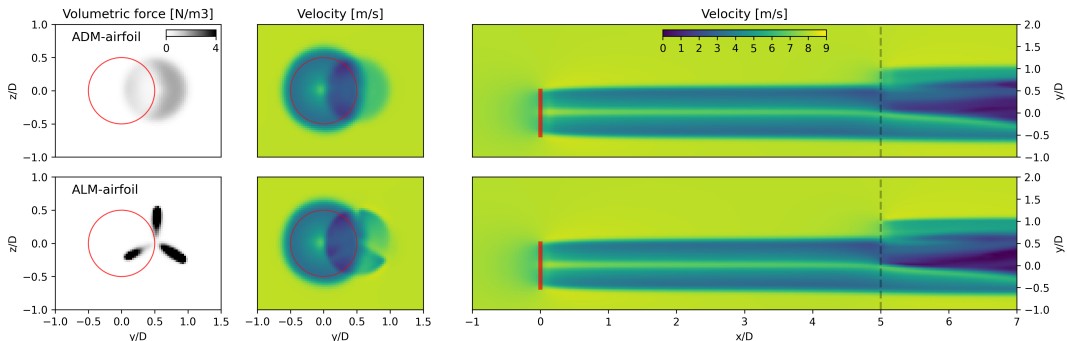

**Figure 6.** Model comparison for two turbines under uniform velocity inflow $U_\infty = 8$ ms$^{-1}$, using ADM-airfoil in WT1 and testing ADM-airfoil and ALM-airfoil in WT2: Instantaneous force (left) and velocity (center) in a vertical plane at WT2 location. The WT1 rotor plane is marked by the red circle. (right) instantaneous velocity in a horizontal plane at hub height, where the WT2 vertical plane position is marked with a dashed line.

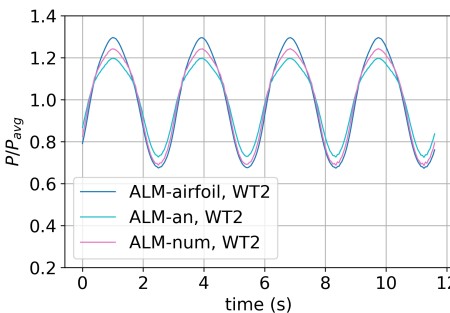

**Figure 7.** ALM comparison of the time varying power output P normalised with the time average power $P_{avg}$ for WT2 being partially waked by WT1. The rotational speed $\Omega$ is fixed in all the models.

## 3.2 ABL inflow

The last step is to compare the model performance in a turbulent neutral ABL inflow condition, which is related to the problem of simulating wind farms in real field conditions. For this inflow condition, the same layout of two turbines as the uniform inflow case is simulated. At hub height a time averaged velocity of $U_\infty = 8$ ms$^{-1}$ and a typical onshore site turbulence intensity, as measured with a cup anemometer, of approximately $TI_{cup} = 9.75\%$ are specified.

As in the previous case, the ADM-airfoil is used to simulate WT1 in order to create the same wake inflow condition for WT2. Also, in WT2 the reference velocity $U_{ref}$ is fixed, a condition that determines the rotation speed $\Omega$. Following the same procedure as for uniform free stream inflow applied to the ABL case, the fixed values for WT2 are $U_{ref} = 6.867$ ms$^{-1}$ with a corresponding $\Omega = 0.823$ $s^{-1}$. It is important to remark that no change is needed in the analytical (ADM-an and ALM-an)





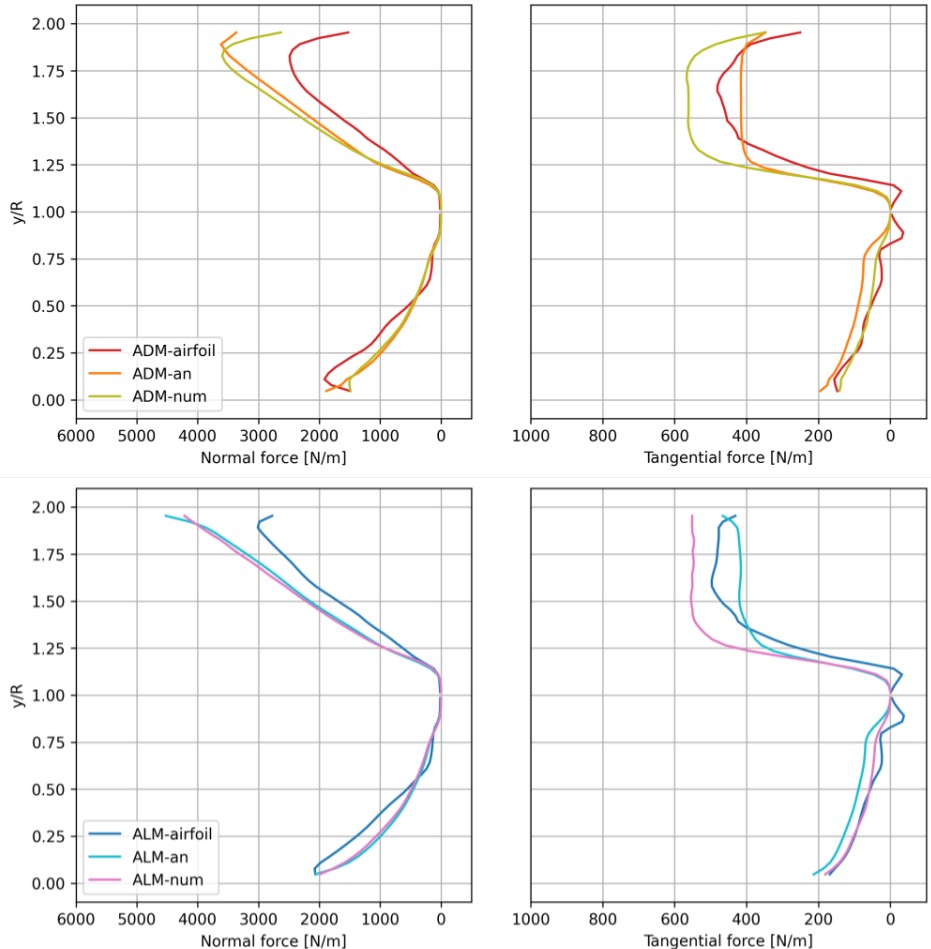

**Figure 8.** Model comparison for two turbines in an uniform velocity inflow $U_\infty = 8 \text{ ms}^{-1}$: time average (left) normal and (right) tangential forces distribution on WT2 for the azimuthal angles $90°$ +/- $10°$ ($y/R$ from 0 to 1) and azimuthal angles $270°$ +/- $10°$ ($y/R$ from 1 to 2).

nor the numerical approaches (ADM-num and ALM-num) to be applied in turbulent inflows. This means that the analytical

relations are still valid as well as the calibration tables obtained for uniform flow.

In Fig. 9 the solutions for ADM-airfoil and ALM-airfoil are presented, showing the instantaneous volumetric force and velocity on a vertical plane normal to the flow at WT2 position, and the velocity on a horizontal plane at hub height. It can be noticed that the local force distribution on the ADM reacts to the non-uniform spatial distribution of the velocity field on the rotor plane. As it can be seen in the instantaneous velocity field, in this complex inflow condition the models need to

respond not only to the partial wake impact but also to the time varying velocity due to of the vertical boundary layer profile. This response is clearly different between the ADM-airfoil and the ALM-airfoil. To visualize this difference, in Fig. 10 the time variation of the power output of both models during a 10-min period is shown. The power output of the ADM-airfoil is


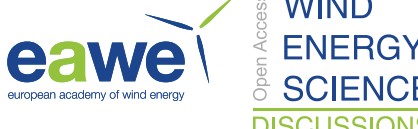

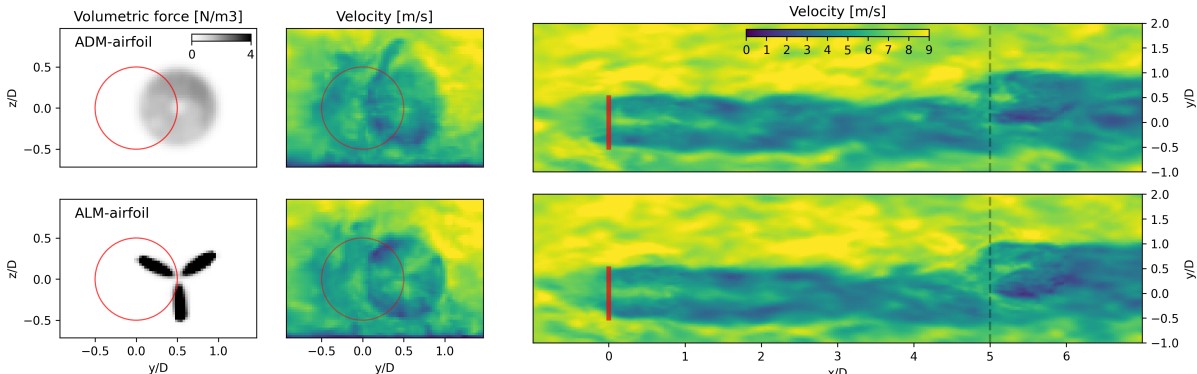

**Figure 9.** Model comparison for two turbines in an ABL with horizontally average velocity inflow at hub height $U_\infty = 8\ \mathrm{m\,s^{-1}}$: Instantaneous force (left) and velocity (right) in a vertical plane at WT2 location. The WT1 rotor position is indicated with a red circle. (right) instantaneous velocity in a horizontal plane position at hub height, with the WT2 vertical plane marked with a dashed line.

only related to the variation of the average velocity over the disc. On the other hand, the ALM-airfoil in addition manages to correctly capture the power variation due to the blades passing through areas of low velocity, including the partial wake and

the non-uniform vertical wind profile. When the three ADM are compared, a good match of the simplified models with the ADM-airfoil is achieved. The major differences are seen when the power reaches local maximums or minimums, registering stronger response in the ALM-num. When the ADM and ALM based on the same formulation are compared, it can be always seen that the ALM can capture additional power variations.

In order to analyze how all the models distribute the forces in this turbulent wake inflow, in Figure 11 the time-averaged

normal $f_n$ and tangential $f_\theta$ forces distribution on WT2 rotor are shown, both for the wake impacted and free side. Due to the ambient turbulence produced by the ABL flow the wake decay and expansion are larger, resulting in a more distributed wake impact over the rotor area and less velocity deficit in the impacted half-side of the rotor. Compared to the uniform free stream inflow case, a similar trend in the forces distribution can be found at both sides of the rotor. Given that in this ABL case the time averaged velocity varies both in the lateral and vertical directions, the forces distribution on the rotor area are shown

in Fig. 12 and 13. In Fig. 12 the normal force distribution over the impacted WT2 rotor for the six models is compared. It is clear how analytical and numerical approaches manage to follow the same trend compared with the airfoil formulation, finding the highest forces values on the upper part of the free side of the rotor and the lowest values in the bottom part of the waked side. Both the analytical and numerical approaches slightly over-estimate values in the free side of the rotor, particularly the ADM-num and ALM-num. In Fig. 13 a similar comparison is shown for the tangential force distribution, finding again the

same overestimation of forces in the free side of the rotor by the ADM-num and ALM-num. It is easy to notice that the force distribution along the radius is less homogeneous than in the uniform inflow case. This problem is due to the local velocity interpolation being affected by the non-aligned position of the rotor area with the mesh, which affects in the same manner to all the models. Later studies would have to analyse this phenomena and how to reduce its effect on the results.



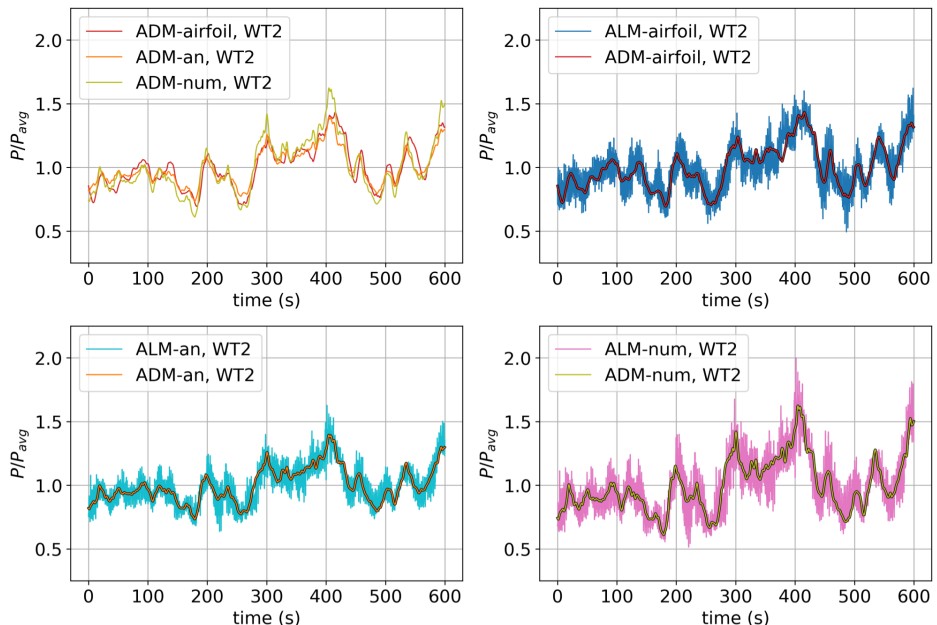

**Figure 10.** Model comparison of the power output time variation normalised with its time average value ($P_{avg}$) for two turbines in ABL with average velocity inflow at hub height $U_\infty = 8\ \mathrm{ms^{-1}}$. (top left) comparison of the three ALM. In the rest of the sub-figures the comparison of the same group of ADM and ALM is shown.

## 4 Conclusions

In this work the previous developments for an ADM with local force calculations based on analytical (Sørensen et al., 2020) or numerical (Navarro Diaz et al., 2019b) approaches are extended to the ALM. These two approaches share the advantage that only limited basic manufacturer WT information is needed, i.e., any combination of the thrust and power coefficient and the tip speed ratio. The ADM and ALM built from these two approaches are compared with the equivalent advanced models based on blade element theory which provides more realistic force distributions but requires information about the operating setting

and the geometric and aerodynamic characteristics of the blade.

When a single turbine facing a uniform free stream inflow is simulated, a close normal and tangential force distribution along the blade is found between all the models. The difficulties to capture the root and tip force distributions obtained with the airfoil data is the major source of differences with the simpler models. The two new ALM have the same distribution as the corresponding ADM, with the only difference being the particular tip correction needed in the ALM.

The models are tested in the challenging case of a WT partially affected by an up-steam wake, both for uniform non-turbulent inflow and turbulent neutral ABL. The analytical and numerical approaches manage to correctly capture the different forces distribution at the different regions of the rotor, with a consistent over estimation of the normal force in the free side and a sub estimation on the waked side. Looking at the tangential force distribution, the numerical approach tends to overestimate the





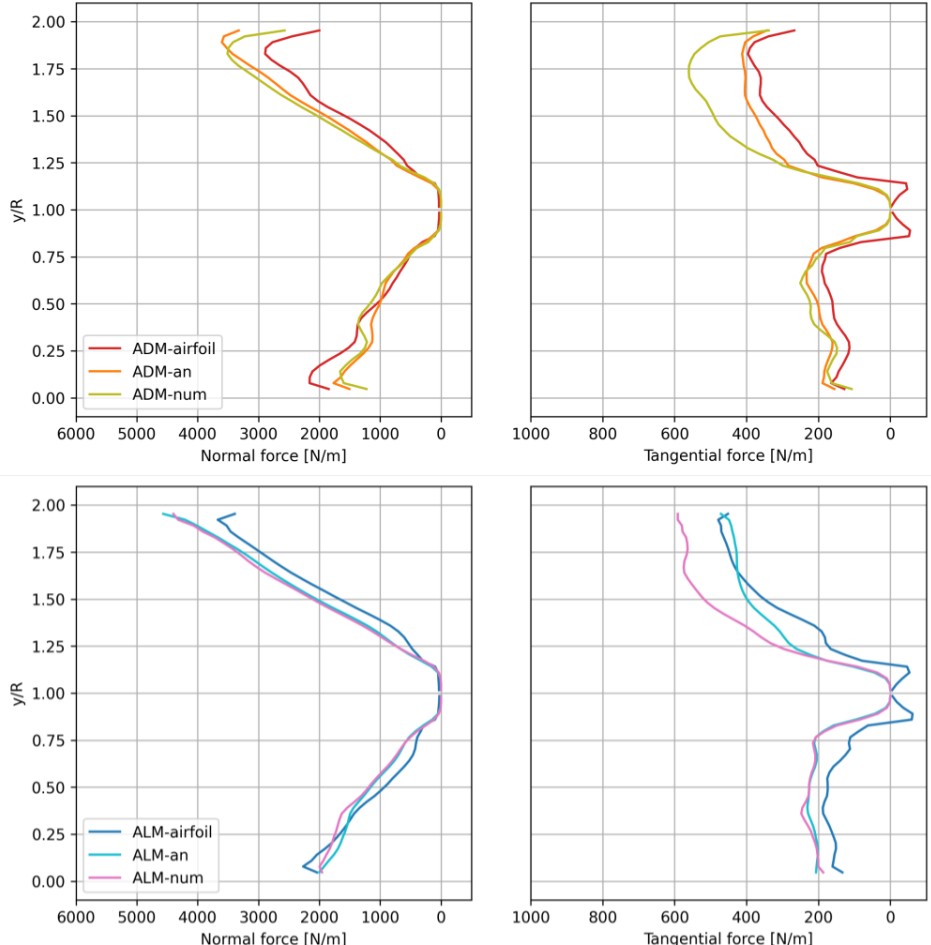

**Figure 11.** Model comparison for two turbines in an ABL average velocity inflow $U_{hub} = 8\,\mathrm{ms}^{-1}$: average (left) normal and (right) tangential forces distribution on WT2 for the azimuthal angles 90 +/- 10 ($y/R$ from 0 to 1) and azimuthal angles 270 +/- 10 ($y/R$ from 1 to 2).

values in the free side. In general, the analytical approach shows a slightly better performance in wake impact cases compared

to the numerical one. The three ALM show the additional capacity of capturing higher frequencies in the power output variation in time. In the uniform inflow case, the clear sign on how the power is reduced when one of the blades passes through the wake region is captured in an equivalent way by the three models. This added feature of the ALM is also visible in the ABL inflow case. The numerical approach has shown higher power fluctuations both in the ADM and ALM. Finally, the extension of both the analytical and numerical approaches from the ADM to the ALM have shown promising results, opening the possibility to

simulate commercial wind farms in transient inflows with ALM without the restriction of private manufacturer blade data.



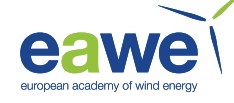


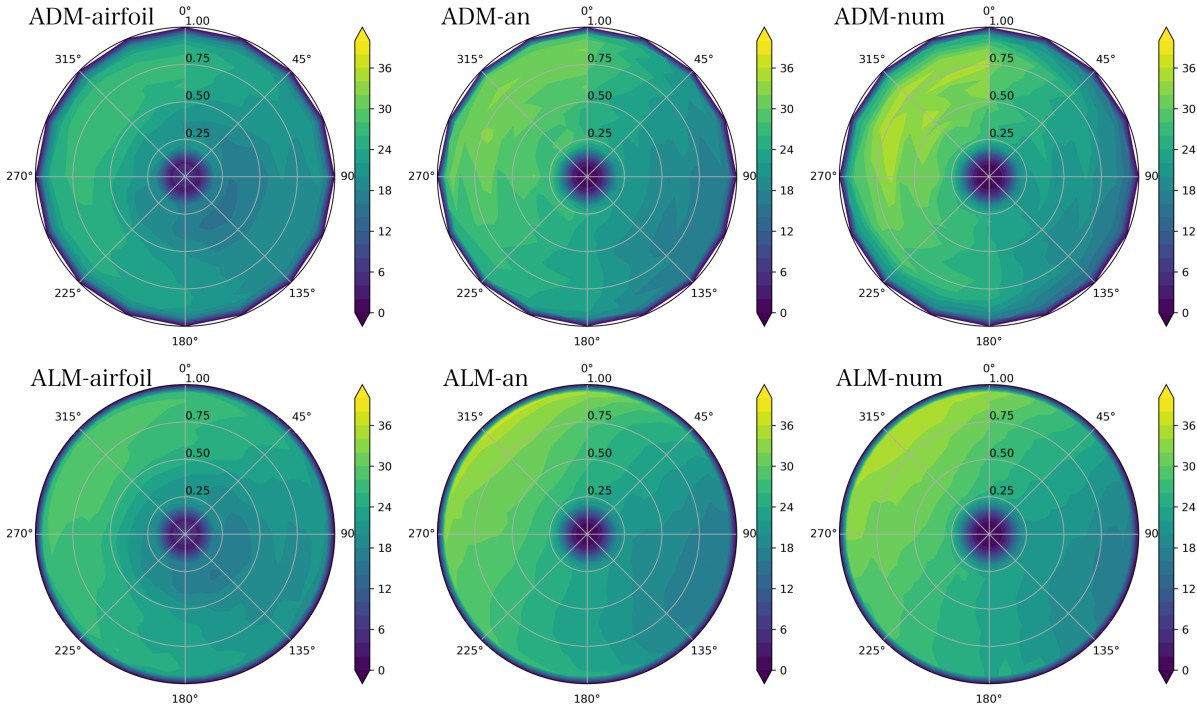

**Figure 12.** Model comparison for the time averaged normal force distribution (N/m2) over the impacted rotor (WT2) for ABL inflow. The wake from WT1 impacts the right side of the rotor plane../Images/figure1.png

## Appendix A: Mesh sensitivity analysis

For this study the well known ADM-airfoil and ALM-airfoil are chosen. The inflow velocity is fixed at $U_{hub} = 8$ ms$^{-1}$ and the tip speed ratio $\lambda = 7.55$ in order to compare the results with the ones published by Asmuth et al. (2020). At this velocity, the rotational speed is approximately $\Omega$= 9.15 rpm = 0.95 $s^{-1}$. These authors carried out a mesh sensitivity study for the 460 ALM-airfoil without tip correction. Particularly, the results for D/32 using the software EllipSys3D are chosen for comparison.

This pair of models are tested for a wide range of mesh resolutions, which are defined by the cell size in relation to the rotor diameter $D$: $D/16$, $D/24$, $D/32$, $D/48$ and $D/64$. It is important to remark that the separation between nodes $\Delta_b$ and $\varepsilon$ are defined in relation to the cell size $\Delta_x$. In the case of the ADM, the number of lines $n_l$ (Eq. 1) also depends on the mesh resolution.

In Fig. A1 it can be seen how the maximum instantaneous vorticity in the near wake changes when the mesh resolution is increased. In the case of the ADM-airfoil, the values of vorticity in the tip and root regions increase when the cell size is reduced. On the other hand, in the ALM-airfoil this pattern is also followed by the development of tip and root helical tubes when the resolution is closer to $D/48$ or higher. The comparison of the vorticity fields between the ADM and ALM and the

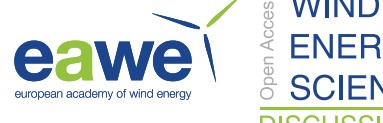

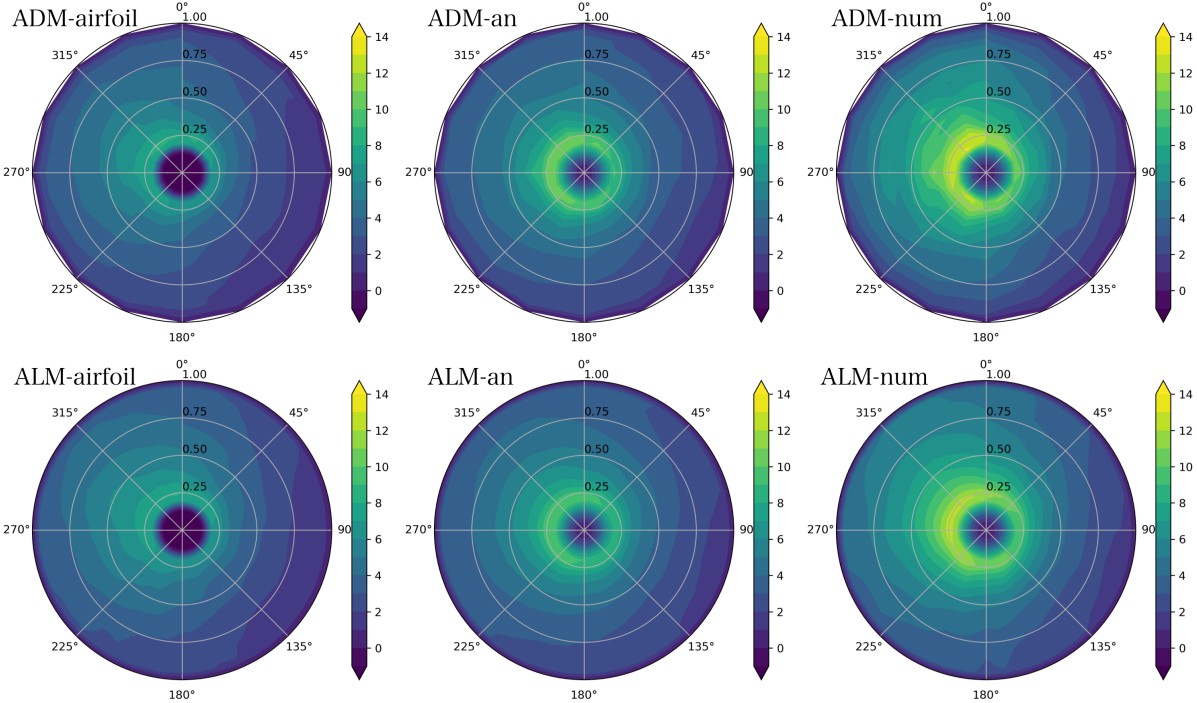

**Figure 13.** Model comparison for the time averaged tangential force distribution (N/m2) over the impacted rotor (WT2) for ABL inflow. The wake from WT1 impacts the right side of the rotor plane.

dependency of the development of helical structures with the mesh resolution was also studied by Martinez et al. (2012), who
arrived at similar conclusions than this sensitivity study.

In Fig. A2 the normal $f_n$ and tangential $f_\theta$ forces distribution along the blade for both models are shown. In the case of the ADM-airfoil, when the resolution is increased the forces near the tip and root regions are reduced, maintaining the values in the rest of the disc. For the ALM-airfoil this pattern is also seen, but with minor changes in the values. A good agreement between the results of Asmuth et al. (2020) and the ones obtained in this work can be seen along the blade. Expected small differences
are found due to the tip correction adopted in this work, where $\varepsilon$ varies depending on the radial position.

In order to see how the forces distribution converge when the mesh is refined, the L2 error for the normal force $f_n$ distribution is calculated for 32 fixed radial positions $j$ along the blade as

$$L2 = \sqrt{\frac{\sum_j (f_{n,j} - f_{n,j,D/64})^2}{\sum_j f_{n,j,D/64}^2}} \tag{A1}$$

where $f_{n,j,D/64}$ is the solution for the higher resolution $D/64$. The same expression is used for the tangential force $f_\theta$. In
Fig. A3 it can be observed how the L2 error is reduced rapidly when the mesh is refined, showing similar values for both models. The error for the normal force is lower compared to the results for the tangential force. For the specific resolution $D/32$, the L2 errors in the normal force are 1.2% and 1.4% for the ADM-airfoil and ALM-airfoil, respectively. In the case of





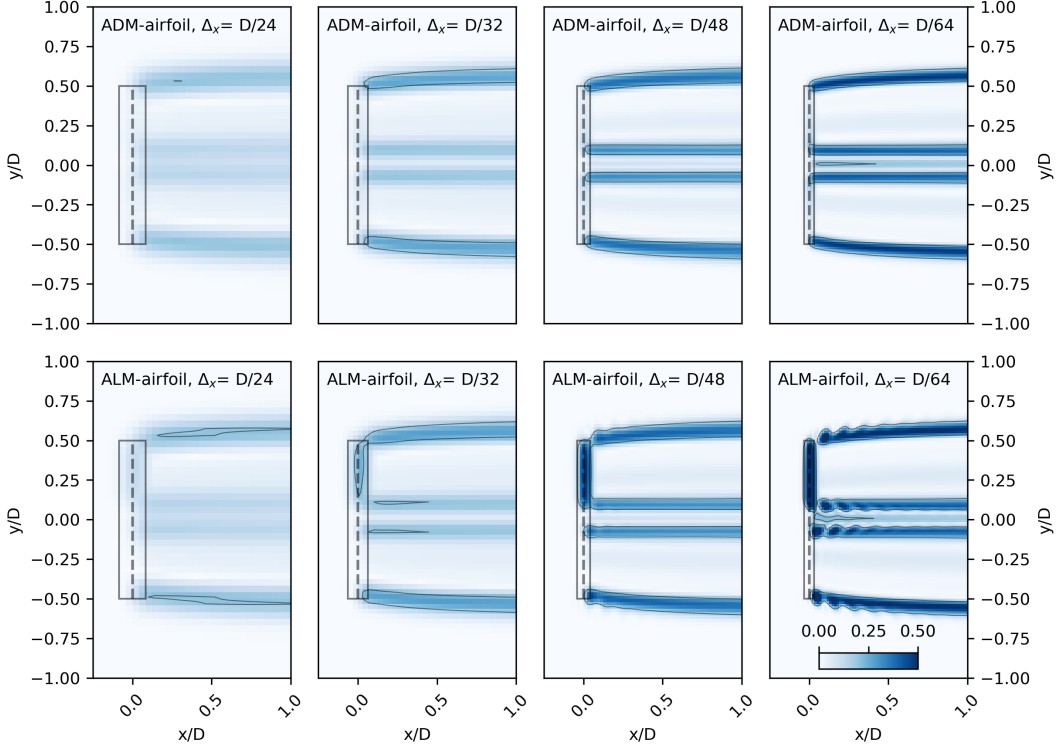

**Figure A1.** Mesh sensitivity study for the ADM-airfoil and ALM-airfoil: instantaneous vorticity on a plane at hub height. Part of the extension of the forces distribution is marked with a rectangle, assigning a total rectangle width of $2\varepsilon$.

the tangential force, the errors are 3.5% and 3.8%. The changes in the forces are reflected in the thrust $T$ and power output $P$, as can be seen also in Fig. A3. In the case of the ADM-airfoil, $T$ and $P$ are reduced with the increasing resolution. This relation

between the reduction of $\varepsilon$, or reduction of the cell size $\Delta_x$, and the decrease of the power was also found by Martinez et al. (2012). When the resolution $D/32$ is compared with the higher resolution, the thrust force is 0.5% higher with the ADM-airfoil and 0.7% with the ALM-airfoil. The power output is 2.0% higher with the ADM-airfoil and 1.7% with the ALM-airfoil.

From the results obtained in this mesh sensitivity study, the resolution $D/32$ has been chosen for the simulations in this work. This choice is a compromise between a low L2 error in the forces distribution and the turbine outputs and the affordable

computational cost for ABL simulations. This cost is especially important in this work, due to the fact that the new ALMs, based on basic manufacture information, are created in order to open the possibility to simulate big wind farms with commercial WTs and long transient ABL problems.



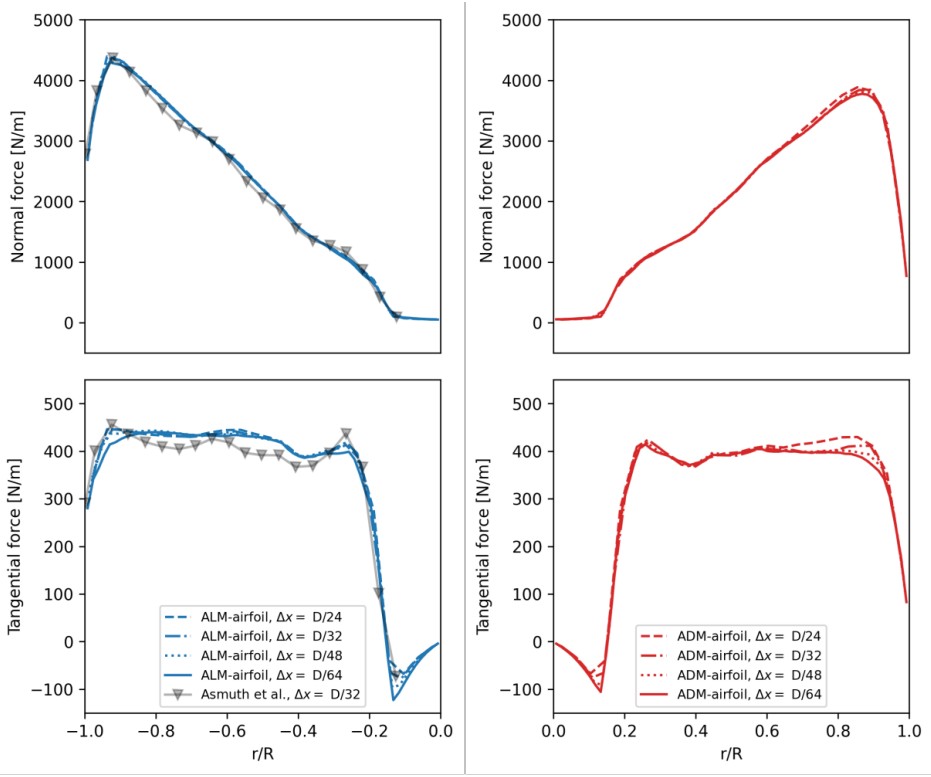

**Figure A2.** Mesh sensitivity study for the ADM-airfoil and ALM-airfoil models: normal $f_n$ (upper) and tangential $f_\theta$ (lower) forces distribution along the blade using different mesh resolutions. The EllipSys LES result from (Asmuth et al., 2020) for the resolution $D/32$ the ALM-airfoil (no tip correction) is also plotted.

*Author contributions.* Gonzalo P. Navarro Diaz developed the model code, performed the simulations, plot the results and wrote the manuscript. Alejandro D. Otero,Henrik Asmuth, Jens NørkærSørensen and Stefan Ivanell contributed with ideas, corrections and modifications both in the choice of models and in the results and text.


*Competing interests.* The authors have the following competing interests: At least one of the (co-)authors is a member of the editorial board of Wind Energy Science.

*Acknowledgements.* The simulations were performed on resources provided by the Swedish National Infrastructure for Computing (SNIC).

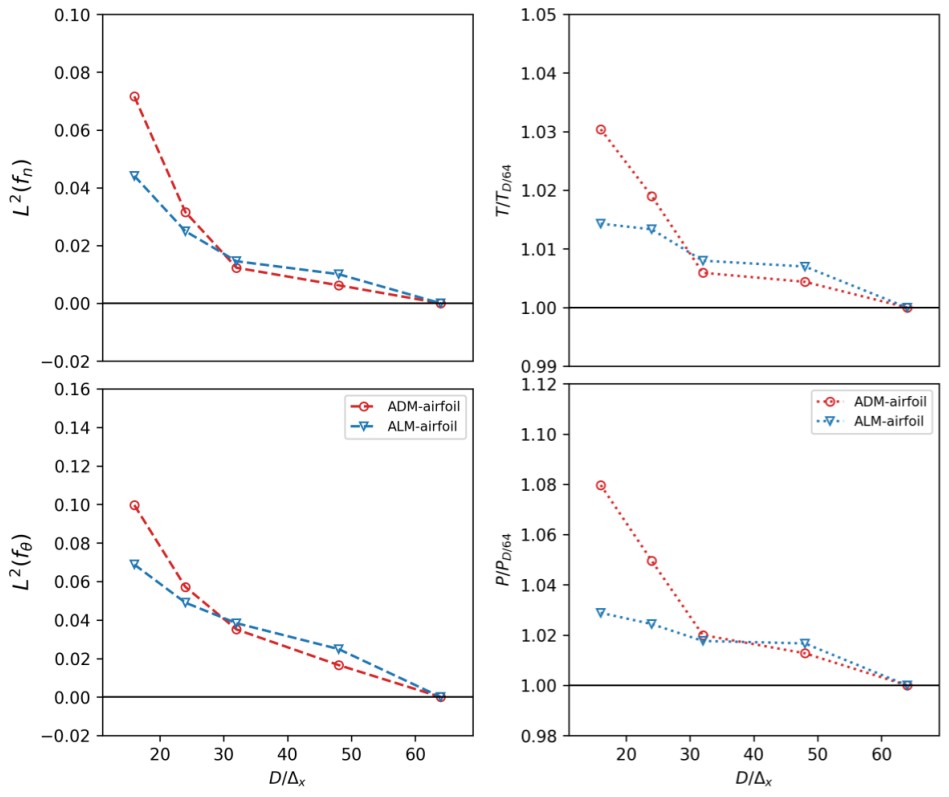

**Figure A3.** Mesh sensitivity study for the ADM-airfoil and ALM-airfoil models: On the left, L2 error for the normal $f_n$ (upper) and tangential $f_\theta$ (lower) force distributions in different mesh resolutions. The error is calculated comparing with the distribution of the higher resolution case. On the right, thrust $T$ (upper) and power $P$ (lower), where the values are divided with the value obtained with the higher resolution.

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
