# Peer review of "Actuator line model using simplified force calculation methods"

_Wind Energy Science, 2022_

## Author Response (AR1)

**Responses to Anonymous Referee 1, 19 Dec 2022**

First of all, the author would like to thank the Reviewer for his/her time and effort reviewing our manuscript.

- **Section 2.2 Numerical set-up**

  1. *Comment from Referee: Figure 2: it would be nice to add some pictures of the grid used for the different flow regions.*

     **Author's response:** Thank you for the suggestion on the figure. The authors think that an extra image of the mesh used would not add additional relevant information to the one presented in a simple way in Figure 2. Particularly for these cases shown, the mesh is very simple, being uniform in the three directions (x, y, z) for each region of refinement. This highlights the main advantage of the actuator line model, in which it is not necessary to include the 3D geometry of the blade since the forces can be distributed over a regular mesh.

**Responses to Anonymous Referee 2, 28 Dec 2022**

First of all, the author would like to thank the Reviewer for his/her time and effort reviewing our manuscript. The answers to your questions have been ordered based on the sections of the manuscript.

- **Section 2.1 Force modeling**

  1. *Comment from Referee: (question 7) Is there a time step limit to the new ALM approaches? Is Eq. 2 still valid here? It is actually the most severe constraint for an ALM.*

     **Author's response:** thanks for the interesting question. The new actuator lines models inherit the time step limitation of the classical actuator line model based on airfoil data. That is why equation 2 is still valid. Other papers have tried to get rid of this strong time step limitation by proposing the actuator sector model (https://onlinelibrary.wiley.com/doi/abs/10.1002/we.1722). The implementation of this model in future works is a very good idea indeed.

  2. *Comment from Referee: (question 9) Can the current approaches be applied to rotor yawing when the main wind direction is not aligned with the rotor orientation?*

     **Author's response:** Thanks for bringing up this interesting question. These new actuator line models have not been tested for yaw conditions. In future works it would be interesting to simulate this condition and compare it with the accurate solution that can be obtained with the classical actuator line model based on airfoil data.

- **Section 2.1.1 Airfoil based forces**

  1. *Comment from Referee: (question 1) Page 7, since $U_{rel}$ is determined by Eq. 13 which uses the local velocity components, why is it necessary to determine the operating region as discussed in the afore paragraph, i.e., "by creating a calibration table with relation between the average velocity over the disc $(< U_d >)$ and the $U_{ref}$"?*

**Author's response:** Please, let me clarify this difference between the two velocities, $U_{rel}$ and $U_{ref}$. To calculate the relative local velocity $U_{rel}$ at each node of the blade using Eq. 13, we can see that it is necessary to know the rotational speed, $\omega$. $\omega$ depends on the reference velocity $U_{ref}$ and it is obtained through the manufacturing curve $\omega$ vs. $U_{ref}$. That is why it is first necessary to know the reference speed $U_{ref}$ to then calculate the local relative speed $U_{rel}$.

2. ***Comment from Referee: (question 8) How is the local velocity $U_{d,i}$ obtained? A lot of previous studies showed that the velocity sampling play an crucial role for ALM. I think it is in SOWFA which is not the topic of this work, but still it will be helpful to mention in the paper.***

   **Author's response:** Thank you for highlighting the need to clarify how the local velocity at each node $U_{d,i}$ is calculated. Now we have extended the text. Despite this, in line 430 we discuss the need to continue investigating interpolation, since when the area of rotation is not perfectly aligned with the mesh this causes small oscillations in the forces along the blade. Pre-calculating the velocity gradient in all rotor cells would be a good solution (as it is done in SOWFA´s actuator line model implementation), but would bring with it a higher computational cost.

   **Author's changes in manuscript:** (line 162) "It is important to clarify that the local velocity vector is calculated by linear interpolation with the 8 closest cells to the node position."

- **Section 2.1.3 Numerical force distribution**

  1. ***Comment from Referee: (question 2) n Eq. 31 and 32, it seems that F' terms are not defined.***

     **Author's response:** The expressions of the forces $\Delta F'_{n,i}$ and $\Delta F'_{\theta,i}$ are expressed in the Eq. 27 and 28. These forces per unit of area does not take into account the tip and root corrections.

  2. ***Comment from Referee: (question 3) Line 241, "a simulation is run for ...": what kind of simulation is it? What method is used in the simulation? Need more details here.***

     **Author's response:** Thank you for finding the need to give more details about the characteristics of these calibration simulations. Now we have extended the text.

     **Author's changes in manuscript:** (line 243) "In each of these simulations, the turbine stands alone in the middle of the domain, following the same detailed specifications that are addressed in the section 2.2 for the mesh sensitivity analysis case".

  3. ***Comment from Referee: (question 4) It is confusing how table 3 is obtained. Line 246 to Line 248 are not very clear and maybe a reorginazation of the sentence is needed. Please use simpler but more straightforward expression. Especially, what are inputs and what are outputs? Are additional simulations needed here? if so, what kind of simulations?***

     **Author's response:** Thank you for noticing the need to clarify the paragraph where the construction of table 3 is explained. We have modified this paragraph as follows:

     **Author's changes in manuscript:** (line 248) "To construct table 3, for a each input values $C_T$, $C_P$ and $\lambda$ in accordance to a reference velocity $U_{ref}$ in the WT operating range, new extra simulations are carried out in which different inflow velocities $U_\infty$ are imposed along with the resulting ADM forces. For each of this simulations, as outputs it is saved for each discrete radial position $r_i$ the local velocity $U_{d,i}$, and forces $\Delta F_{n,i}$ and $\Delta F_{\theta,i}$, columns 5 to 7 of table 3."

4. ***Comment from Referee: (question 5) It seems that for the numerical approach those tables have to be remade whenever a different rotor is used. WIll it be too complicated for a real implementation? What are the overhead of running those preprocessing simulations to generate tables?***

   **Author's response:** This is a very interesting question for this type of actuator model that needs a previous calibration table calculated. As has been done before for actuator disc models (see for example [1]), it is very common to use this procedure when only Ct and no airfoil data are available to calculate the forces. To create these tables it is necessary to carry out several previous simulations where the turbine is individually simulated facing different input speeds. Although it is necessary to create tables for each type of turbine, its computational cost is very low compared to simulating the entire wind farm.

5. ***Comment from Referee: (question 6) It might be helpful for readers to follow if flowcharts are given for both the analytical and numerical approaches.***

   **Author's response:** Thanks for the suggestion. We made a special effort to explain step by step all the necessary calculations to obtain the forces. That's why we think flowcharts would just repeat the text without adding new information.

**References**

[1] M. P. van der Laan, N. N. Sørensen, P.-E. Réthoré, J. Mann, M. C. Kelly, N. Troldborg, J. G. Schepers, and E. Machefaux. An improved $k$ - $\varepsilon$ model applied to a wind turbine wake in atmospheric turbulence. *Wind Energy*, 18(5):889–907, 2015.